# A Novel Hypoxia-Immune Signature for Gastric Cancer Prognosis and Immunotherapy: Insights from Bulk and Single-Cell RNA-Seq

**DOI:** 10.3390/cimb47070552

**Published:** 2025-07-16

**Authors:** Mai Hanh Nguyen, Hoang Dang Khoa Ta, Doan Phuong Quy Nguyen, Viet Huan Le, Nguyen Quoc Khanh Le

**Affiliations:** 1International Ph.D. Program in Cell Therapy and Regenerative Medicine, College of Medicine, Taipei Medical University, Taipei 110, Taiwan; nguyenmaihanh@vmmu.edu.vn; 2Pathology and Forensic Medicine Department, 103 Military Hospital, Hanoi 151000, Vietnam; 3School of Computer Science, Duy Tan University, Da Nang 550000, Vietnam; tahdangkhoa@duytan.edu.vn; 4DTU AI and Data Science Hub (DAIDASH), Duy Tan University, Da Nang 550000, Vietnam; 5Institute of Biomedicine, Hue University of Medicine and Pharmacy, Hue University, Hue City 49120, Vietnam; ndpquy@hueuni.edu.vn; 6Vietnam Department of Medical Genetics, Hue University of Medicine and Pharmacy, Hue University, Hue City 49120, Vietnam; 7Department of Thoracic Surgery, Khanh Hoa General Hospital, Nha Trang 650000, Vietnam; huanleviet.dr@gmail.com; 8In-Service Master Program in Artificial Intelligence in Medicine, College of Medicine, Taipei Medical University, Taipei 110, Taiwan; 9AIBioMed Research Group, Taipei Medical University, Taipei 110, Taiwan; 10Translational Imaging Research Center, Taipei Medical University Hospital, Taipei 110, Taiwan

**Keywords:** gastric cancer, computational biology, gene expression, hypoxia, immunotherapy, prognosis

## Abstract

**Background**: Hypoxia and immune components significantly shape the tumor microenvironment and influence prognosis and immunotherapy response in gastric cancer (GC). This study aimed to develop hypoxia- and immune-related gene signatures for prognostic evaluation in GC. **Methods**: Transcriptomic data from TCGA-STAD were integrated with hypoxia- and immune-related genes from InnateDB and MSigDB. A prognostic gene signature was constructed using Cox regression analyses and validated on an independent GSE84437 cohort and single-cell RNA dataset. We further analyzed immune cell infiltration, molecular characteristics of different risk groups, and their association with immunotherapy response. Single-cell RNA-seq data from the TISCH database were used to explore gene expression patterns across cell types. **Results**: Five genes (*TGFB3*, *INHA*, *SERPINE1*, *GPC3*, *SRPX*) were identified. The risk score effectively stratified patients by prognosis, with the high-risk group showing lower overall survival and lower T-cell expression. The gene signature had an association with immune suppression, *ARID1A* mutation, EMT features, and poorer response to immunotherapy. Gene signature, especially *SRPX* was enriched in fibroblasts. **Conclusions**: We developed a robust hypoxia- and immune-related gene signature that predicts prognosis and may help guide immunotherapy strategies for GC patients.

## 1. Introduction

Gastric cancer (GC) is the fifth most prevalent cancer globally and the fourth most prominent contributor to cancer-related deaths worldwide, associated with a challenging prognosis [1,2]. The prognosis for overall survival (OS) in GC patients is poor, with individuals suffering from advanced stages typically having a median OS of about one year [3]. GC also is a heterogenous tumor so individual outcomes depend on various factors due to the tumor microenvironment (TME). Currently, there are numerous treatment options available for GC, including surgery, chemotherapy, radiotherapy, targeted therapies, and especially immunotherapy [4]. However, given the high cost of immunotherapy, particularly in low-resource settings, its limited effectiveness in some patients remains a significant concern. The underlying reasons for this are still being studied and clarified [5]. Therefore, it is essential to identify new biomarkers that can effectively predict the prognosis and immunotherapy response of GC patients.

In GC and other solid tumors, hypoxia is recognized as a critical component of TME, playing a direct role in promoting malignant characteristics, including tumor progression, invasion, and metastasis [6,7,8,9,10]. It also could regulate the occurrence of GC by angiogenesis, proliferation, metastasis, apoptosis, and stemness of GC cells [11]. Over the past decade, the emergence of immunotherapy has provided a novel treatment strategy for advanced GC patients with favorable clinical benefits due to some insights into TME [12,13]. Immune cells have a crucial role in many metabolic pathways resulting cancer cells [14]. In addition, immune-related genes contribute to activating and mobilizing immune cells, releasing inflammatory factors, and significantly impacting cancer development [15]. Understanding how immune cells and cancer cells interact is key to figuring out the complexities of tumor growth and spread, and the reasons why some patients develop resistance to immunotherapy. Therefore, the identification of significant genes for prognosis and immunotherapy in patients is crucial. Several other studies have identified genes associated with either immune or hypoxia status, but no research has yet investigated genes related to both conditions that predict the response to immunotherapy in GC [16,17]. Our study developed a novel gene signature that combined both immune and hypoxia factors to predict outcomes and immunotherapy response in GC patients. We analyzed its association with clinicopathologic and molecular characteristics, including immune cell infiltration and single-cell profiles, aiming to advance personalized medicine for GC.

## 2. Materials and Methods

### 2.1. Patients’ Cohort and Preparation

We gathered 448 samples from The Cancer Genome Atlas (TCGA) data portal (https://portal.gdc.cancer.gov/ (accessed on 1 August 2023)), consisting of 412 tumor specimens and 36 normal specimens. Out of these, 404 samples with comprehensive transcriptomic data, survival information and clinical details (age, gender, N stage, T stage, the stage of the patients, etc.) were utilized to develop a risk model. External validation of the data was conducted using an independent cohort. The expression data of 433 samples corresponding to clinical information of GSE84437 from Gene Expression Omnibus (GEO) database (https://www.ncbi.nlm.nih.gov/geo (accessed on 1 August 2023)) were downloaded. In addition, a set of 4677 genes related to the immune system was obtained from InnateDB (https://www.innateDBdb.com/ (accessed on 1 August 2023)) databases [18] accompanied by the acquisition of 200 genes from the hallmark gene sets linked to hypoxia available in the Molecular Signatures Database (MSigDB version 6.0) [19].

### 2.2. Identification of Hypoxia- and Immune-Related Genes and Development Risk Signature

The R package “DESeq2” (version 1.30.0) was employed to identify differentially expressed genes (DEGs) between 412 tumor and 36 non-tumor samples, using criteria of |log2(Fold Change)| > 0.5 and a false discovery rate (FDR) < 0.05. Subsequently, hypoxia-immune-related genes were extracted by intersecting three gene sets: hypoxia-related genes, immune-related genes, and DEGs. To visualize these results, an online tool was used to create a Venn diagram (https://bioinformatics.psb.ugent.be/webtools/Venn/ (accessed on 1 August 2023)) [20]. For a deeper investigation into gene function within hypoxia and immune-related genes, we conducted functional enrichment analyses using Kyoto Encyclopedia of Genes and Genomes (KEGG), Gene Ontology (GO), and protein–protein interaction (PPI) networks. Enrichment of GO terms and KEGG signaling pathways was assessed based on the criterion of FDR < 0.05.

To identify hypoxia-immune-related genes with valuable prognostic significance, we employed the survival package in R [21]. Our approach conducted univariate Cox regression analysis to thoroughly investigate the correlation between the expression levels of hypoxia-immune-related genes and the overall survival (OS) outcomes of each patients [22]. Genes showing significant prognostic value (*p* < 0.05) were further selected using stepwise multivariate Cox regression modeling. By assigning weights based on their prognostic value, we created a risk signature that can help guide treatment decisions and improve patient care. The formula used in the Cox model to calculate the risk score for each patient was as follows:Risk score = Σ[Expression level of each Gene × coefficient]

The patients then were categorized into low- and high-risk groups using the median value of the risk score to evaluate and confirm the predictive accuracy of risk signature. The R package “survival” was used to create Kaplan–Meier (K–M) curves through the log-rank test, allowing for a detailed comparison of survival rates and patterns between the low- and high-risk groups. Additionally, time-dependent receiver operating characteristic (ROC) curve analysis was conducted using the R package “survivalROC”. The analysis spanned 1-, 3-, and 5-year survival periods and aimed to illustrate the sensitivity and specificity of the risk signature [23]. In addition to the analysis, the gene signature’s prediction was further validated in the GSE84437 dataset, providing more evidence to support the findings.

### 2.3. Identification of the Relationship Between Risk Scores and Clinical Factors

To assess the relationship between risk scores and other prognostic biomarkers, the correlation between the risk score and traditional clinical factors was analyzed. Additionally, univariate and multivariate Cox regression analyses were performed on validation datasets to determine the prognostic significance of the risk score and other clinical factors, aiming to establish whether the risk score is an independent prognostic factor. We carefully developed an effective and informative nomogram to help clinical practitioners predict the survival risk of GC patients. This nomogram integrated risk scores and clinical parameters, providing a comprehensive and quantitative analytical tool. The calibration curves, constructed using the robust R package “rms” [24] are pivotal in evaluating the predictive precision of the nomograms.

### 2.4. In-Depth Analysis of Molecular and Immunological Features in Different Risk Score Groups

The TCGA database provided open-source whole exome sequencing (WES) data for Stomach Adenocarcinoma (STAD). Mutation annotation format (MAF) files, which included clinical data for all patients, were obtained. Using the Illumina HiSeq 2000 (Illumina Inc., San Diego, CA, USA) Whole Exome Sequencing platform, sequencing data from all TCGA-STAD patients were analyzed and classified into high-risk and low-risk groups based on gene signatures. MAF files were processed using the R packages “maftools” [25] and “TCGA biolinks” [26].

To further explore the association between the risk score and the immunological microenvironment, the Immuno-Oncology Biological Research (IOBR) approach was employed. Using the CIBERSORT (https://cibersort.stanford.edu/ (accessed on 1 August 2023)) method, the relative proportions of 22 different types of infiltrating immune cells were estimated for each patient. Additionally, user-defined signatures associated with the immune microenvironment were evaluated through single-sample Gene Set Enrichment Analysis (ssGSEA), Principal Component Analysis (PCA), and Z-score methods. Lastly, the relationships between risk score, immune cell infiltration, and immune-associated markers were thoroughly investigated.

### 2.5. The Ability of Gene Signatures to Predict Immunotherapy Response

To investigate the ability of predicting the immunotherapy response of signature, we used two different score systems. Immunophenoscore (IPS) is a reliable predictor of Immune Checkpoint Inhibitor (ICI) response, calculated from the expression levels of key components involved in tumor immunity such as immunoregulatory factors, MHC molecules, suppressor cells and effector cells, etc. Ranging from 0 to 10, the IPS score is determined by z-scores representing gene expression of specific cell types. The IPS scores for each GC patient were sourced from The Cancer Immunome Atlas (TCIA) (https://tcia.at/home/ (accessed on 1 August 2023)) [27]. In addition, Tumor Immune Dysfunction and Exclusion (TIDE) score was utilized to assess the effectiveness of immunotherapy. This score combines T cell exclusion from cold tumors and T cell dysfunction from hot tumors, giving a comprehensive evaluation of the immune response in cancer patients. To obtain the TIDE score, we used an online tool (http://tide.dfci.harvard.edu/ (accessed on 1 August 2023)) [28] that has been widely recognized and validated and then performed a time-dependent ROC analysis using the “survivalROC” package in R to evaluate the prognostic significance of the TIDE score. By assessing areas under the curve (AUC), we compared the predictive performance of the risk score and TIDE, providing insights into their prognostic capabilities.

### 2.6. Single Cell RNA-Sequencing (scRNA-Seq) Analysis of Gene Signature

The expression of gene signature was analyzed and visualized using the GSE134520 and GSE167297 scRNA-seq dataset from the Tumor Immune Single-cell Hub (TISCH) database (http://tisch.comp-genomics.org/ (accessed on 1 August 2023)). Signature genes were sourced from the Hallmark database, and their expression levels were compared across immune cells, malignant cells, and stromal cells. In total, cells were categorized into 9 major cell types. The Harmony algorithm was employed to integrate three distinct scRNA-seq datasets, which were thoroughly analyzed following the Seurat pipeline. Each cell cluster was annotated using the “SingleR” R package [29] and supported by literature references.

### 2.7. Statistical Analysis

In our study, we used Cox regression analysis to examine survival outcomes. This technique provided hazard ratios (HRs) and confidence intervals (CIs) to understand the relationship between variables. To assess the impact of individual variables on survival rates, we employed Kaplan–Meier analysis and the log-rank test. This allowed us to compare survival curves between groups and determine statistical significance. We also compared continuous variables using *t*-tests to identify any significant differences across risk groups. Furthermore, we conducted chi-square tests to examine variations in clinical attributes between risk score groups. This helped us understand associations between risk scores and specific clinical characteristics. Statistical significance was determined using a two-sided *p*-value threshold of less than 0.05. All analyses were performed using R-v4.1.1, a widely recognized statistical software.

## 3. Results

### 3.1. Development and Validation of Hypoxia- and Immune-Related Gene Signatures

Following a comprehensive analysis of 412 malignant and 36 non-malignant samples from the TCGA cohort, a detailed investigation identified 19,685 DEGs. Within this set, 13,351 genes demonstrated up-regulation, while 6334 genes displayed down-regulation in malignant samples compared to their non-tumor counterparts. (Figure 1a). These DEGs then intersected with immune-related genes from InnateDB and hypoxia-related genes from MSigDB, yielding a total of 44 DEGs links to both hypoxia and immune status (Figure 1b). To better understand the functional roles of these 44-hypoxia-immune-related genes, we performed GO annotation and KEGG pathway enrichment analyses. This detailed investigation revealed that the “HIF-1 signaling pathway” is closely associated with the functions of the 44 identified DEGs, highlighting the molecular complexities involved. Furthermore, the most significantly enriched terms across biological processes (BP), molecular functions (MF), and cellular components (CC) were unveiled as “Regulation of cell death”, “Protein-containing complex binding”, and “Collagen-containing extracellular matrix”, respectively. The PPI network of the 44 hypoxia-immune-related genes was meticulously visualized, adding another layer of understanding to the molecular interplay (Appendix A).

In the TCGA-STAD dataset, univariate Cox regression analysis identified five genes (*TGFB3*, *INHA*, *SERPINE1*, *SRPX*, and *GPC3*), associated with both immune and hypoxia status, as being significantly linked to patient overall survival (*p* < 0.05 and HRs > 1) (Appendix A). A prognostic gene signature was formulated through multivariate Cox regression analysis, involving the calculation of coefficients (Appendix A). Afterward, the risk score was formulated using the following equation:Risk score = [Expression level of *TGFB3* × 1.906 × 10^−5^] + [Expression level of *INHA* × 3.365 × 10^−4^] + [Expression level of *SERPINE1* × 7.965 × 10^−6^] + [Expression level of *GPC3* × 6.999 × 10^−6^] + [Expression level of *SRPX* × 6.395 × 10^−5^].

After dividing patients based on median risk scores, The K–M analysis showed that GC patients with high-risk scores have shorter overall survival compared to those with low-risk scores (log-rank test, *p* < 0.001) (Figure 1c). Subsequently, we evaluated the predictive capacity of the gene signatures through the utilization of time-dependent ROC curves. The area under the curve (AUC) scores for survival at 1, 3, and 5 years were 0.603, 0.620, and 0.707, respectively, demonstrating a steady improvement in predictive performance over time (Figure 1d). These findings highlight the gene signature’s ability in predicting outcomes of each GC individual.

To confirm the robustness of the established signature, a validation test was conducted using the GSE84437 cohort. Kaplan–Meier analysis yielded results consistent with those from the TCGA cohort, showing that patients classified as low-risk had significantly better survival outcomes compared to high-risk patients (log-rank test, *p* < 0.001) (Figure 2a). Similarly, ROC curves were generated to predict survival at 1, 3, and 5 years based on patient OS, yielding AUC of 0.553, 0.589, and 0.612 for these respective time points (Figure 2b).

### 3.2. Relationship Between Gene Signature and Clinicopathological Parameters

In both the complete TCGA-STAD and GSE84437 cohorts, Cox proportional hazards regression analysis was employed to establish the independence of the risk score in predicting prognosis (Figure 1e and Figure 2c). The results in Table 1 show a strong connection between the gene signature and OS of GC patients, confirmed by both univariate and multivariate analyses (*p* < 0.001). This emphasizes the gene signature’s function as an independent prognostic marker in GC patients, demonstrating its potential significance in clinical practice.

In our study, we analyzed the connection between risk scores and clinicopathological parameters, finding a strong and consistent link between risk scores and the T stage in GC. However, we did not find any notable correlations between the risk score and age or gender factors (Table 2, Figure 1i and Figure 2f). This suggests that the risk score may not be influenced by these demographic factors and emphasizes the importance of considering other clinical parameters when assessing prognosis in GC patients. Additionally, the risk score exhibited higher predictive accuracy for 1, 3, and 5-year survival compared to other clinicopathological parameters (Figure 1f and Appendix A). Integrating risk scores and specific clinicopathological factors enabled us to develop a prognostic nomogram, aiming to provide potential tools that can assist clinicians for predicting individual patients’ survival risk (Figure 1g and Figure 2d). Notably, the calibration curves of the prognostic nomogram demonstrated a high degree of agreement between the predicted and actual 1-, 3-, and 5-year survival rates across the entire TCGA cohort (Figure 1h and Figure 2e). This finding emphasizes the accuracy and reliability of the nomogram as a valuable tool for clinicians in evaluating GC prognosis and guiding treatment decisions.

### 3.3. Microenvironment of Different Risk-Score Groups

To gain deeper biological insights into GC tumorigenesis, we investigated the immunological characteristics of the risk score groups by analyzing gene mutations. Figure 3c presents an overview of the mutation discovered in TCGA-STAD cohort, with top 10 genes with the highest mutation frequency in the high and low-risk groups that are displayed. It is shown that the top five genes (*ARID1A*, *MYCBP2*, *AFF2*, *C3*, and *CNTN4*) are common and had a significantly higher rate in high-risk patients compared to others in low-risk group, which could be contributing to their increased risk of disease progression and poorer outcomes. Additionally, the patients with high-risk score had strong association with some gene mutations such as *TEP1*, *MYCBP2*, and *PCDHGA12* (Figure 3b). Using the CIBERSORT tool, we analyzed the differences in TME between the two risk score groups. This method allowed us to evaluate the presence of 22 types of immune cells within the tumor samples, providing insights into the complex immune landscape. Comparing the high-risk and low-risk cohorts, our result indicated marked variations within the TME. The high-risk group showed lower levels of T cell CD8, T cell CD4 memory activated, T cell follicular, and T cell regulatory Tregs compared to the low-risk group (Figure 3c) (*p* < 0.05). Additionally, we observed contrasting levels of B cell naïve, monocyte, macrophage M2, and resting mast cells between the two risk groups. Notably, these immune cell types were found to be significantly higher in the high-risk group. This suggests an impaired immune response in the high-risk group, which may impact disease progression and treatment outcomes.

We then discovered a robust correlation between risk score values and the infiltration of immune cells. Our results consistently demonstrated a potential relationship between the risk score and the presence of T and B cells within the tumor (*p* < 0.05) (Figure 3d). This suggested that the risk score significantly influences immune infiltration during tumor progression, providing valuable insights into the immune response in GC.

Among the high-risk and low-risk groups, we found that there are differences in TME activity, immune suppression, fibroblast involvement, and metabolic dysregulation. The high-risk group showed a more aggressive tumor microenvironment with greater fibroblast activity, Epithelial–Mesenchymal Transition (EMT), and immune suppression such as MDSCs and specific CAF subtypes, while the low-risk group retained features of non-malignant tissue, including lower fibroblast infiltration and reduced EMT activation (Figure 3e). These findings indicated that high-risk patients had poor prognosis compared to low-risk patients.

### 3.4. The Ability to Predict Immunotherapy Response of Gene Signature

We employed TIDE to evaluate the potential clinical effectiveness of immunotherapy in two groups categorized by risk scores. A higher TIDE score indicates a greater likelihood of immune evasion, suggesting that individuals with higher scores might experience reduced benefits ICI therapy [28]. Our investigation within the TCGA cohort revealed that patients categorized as low risk exhibited lower TIDE scores, MSI score compared to those in the high-risk group (Figure 4a), suggesting that individuals with a low-risk profile might potentially gain more advantages from ICIs therapy compared to high-risk group. Additionally, when comparing the predictive capacity of the risk score to TIDE scores, we observed that the risk score demonstrated greater reliability (Figure 4c). This finding highlighted the potential of the risk score as a robust biomarker for accurately predicting the efficacy of immunotherapy.

We also analyzed the correlation between the risk score and immune cell expression profiles, particularly the IPS, in GC patients. This assessment helped validate the risk score’s effectiveness in predicting responses to ICIs, which was crucial for personalized treatment. Since previous research has highlighted the importance of IPS in forecasting ICI responses, our examination aimed to provide further evidence and enhance the risk score’s reliability as a prognostic tool in immunotherapy for GC patients [27]. Hence, to assess the potential utility of ICIs, we employed the expressions of IPS-CTLA4+PD-1 positive, IPS-CTLA4-negative+PD1-positive, IPS-PD1 negative+CTLA4 positive, and IPS-PD1+CTLA4 negative to evaluate the immunogenicity within different risk groups [30]. Interestingly, our findings revealed that all these scores were higher in the low-risk group, suggesting enhanced responsiveness to ICIs within this group (Figure 4b).

### 3.5. Gene Signature and Tumor Biology

To deepen our understanding of tumor biology, particularly in the context of metastasis and immune evasion, we conducted a comparative analysis of several EMT-related gene expression signatures between high-risk and low-risk groups. Our findings revealed that the signature scores for EMT1, EMT2, EMT3, and Pan-Fibroblast TGF-β response signature (Pan_F_TBRs) were significantly elevated in the high-risk group compared to the low-risk group (*p* < 0.05), suggesting a higher level of EMT activity among high-risk individuals (Figure 4d). Conversely, no statistically significant difference was observed for the WNT target marker between the two groups. Immune microenvironment signatures, including G PAGs, Pan_F_TBRs, Myeloid-derived suppressor cells (MDSCs), and Tumor-associated macrophages (TAMs), also showed significant differences between high-risk and low-risk groups. High-risk patients demonstrated increased scores, suggesting a more immunosuppressive or tumor-promoting environment (Figure 4e). Increased fibroblast (Pan_F_TBRs), MDSC, and TAM activity in high-risk patients may contribute to cancer progression and be associated with poor prognosis.

### 3.6. Single-Cell Profiling of Gene Signature

After examining the distribution of nine different cell types we also found that hypoxia-immune-related gene signature was highly enriched in fibroblast compared to other cell types (Figure 5a,b and Appendix A). This suggested that fibroblasts are not only prominent within the tumor but are also likely to respond or influencing the hypoxic and immune conditions of the tissue. Among the five gene signatures, only the *SRPX* gene showed high expression in fibroblasts (Figure 5e–i). Additionally, we explored the relationship of different cell types with immunologic pathways and oncogenic pathways. Fibroblast and epithelial cells appeared to have the strongest up-regulations of oncogenic pathways (Figure 5d). Fibroblast also showed notable up-regulation of several immune gene sets (Figure 5c), indicating this cell is significantly involved in immune-related processes, possibly contributing to immune suppression within TME.

## 4. Discussion

GC stands as one of the most aggressive malignancies, marked by a high incidence rate and notably low overall survival rates among patients [31]. Recent evidence increasingly indicates the intimate association of hypoxia and immune status with tumor development, cancer progression, and poor survival outcomes in GC patients [32,33]. Currently, immunotherapy is acknowledged as a treatment for various cancer types, including GC. Given the generally low response rates and the high cost, it becomes crucial to identify which GC patients could benefit from ICI. Identifying a specific biomarker is crucial for predicting prognosis and the immunotherapy response, helping clinicians to selectively and strategically include GC patients who are suitable for ICI-based treatments.

In this research, we combined the transcriptomic and clinical data of TCGA-STAD and GSE84437 datasets to develop and validate the gene signature. We screened DEGs between 412 tumors and 36 normal samples in TCGA-STAD dataset. After interesting with immune gene list and hypoxia gene list, we discovered 44 hypoxia-immune-related genes, including 44 genes significantly associated with hypoxia and immune pathways, particularly enriched in ‘regulation of cell death’ and ‘HIF-1 signaling pathway’ biological processes”. These findings align with the previous literature [34,35,36] demonstrating that under hypoxic conditions, there is a reduction in the immunogenic cell death of cancer cells. Furthermore, HIF-1alpha, recognized for its pivotal role in immunity and inflammation, emerges as a key regulator influencing immune cell functionality.

Next, through univariate and multivariate Cox regression modeling, five genes out of the 44 identified- *TGFB3*, *INHA*, *SERPINE1*, *GPC3*, and *SRPX*—were selected for the gene signature. Subsequently, employing weighted gene expression levels, we calculated the risk score and divided patients into distinct risk score groups. Across both TCGA and GEO datasets, this gene signature emerged as a significant biomarker for predicting GC prognosis, persistently demonstrating lower survival outcomes in low-risk populations compared to those at high risk. This aligned consistently with previous research findings [37]. We successfully created a nomogram that integrates the risk score with additional clinicopathological factors aimed at refining prognostic accuracy. This tool assisted clinicians with the ability to predict individual GC patient survival risks. Dai et al. [30] similarly integrated clinicopathological factors and gene signatures into a nomogram, enhancing accuracy in predicting clinical outcomes for hepatocellular carcinoma patients.

Numerous studies highlight the immune system as a key player in combating cancer cells, thus aiding the success of immunotherapy. However, conflicting findings have arisen concerning the specific role of immune cells in determining the prognosis of GC [38]. For a deeper understanding of immune characteristics in GC, our exploration focused on analyzing the differences in immune cell profiles between two groups with varying risk scores. Employing the CIBERSORT tool, we assessed the expression of 22 immune cell types in every single GC specimen. Our evaluation focused on examining how the risk score might mirror the microenvironment in GC, specifically delving into the prognostic relevance of different immune cell types. Some studies have indicated that CD8+ cells play a pivotal role as immune cells in tumor elimination [39,40], while CD4+ T cells contribute to the immune response and are linked to improved prognosis [41]. Consistent with previous research, our findings demonstrate a positive correlation between increased levels of activated CD4 memory T cells and CD8 T cell infiltration with better prognostic outcomes in patients. This observation highlighted the potential of our identified signature to precisely capture the complexity of TME in GC.

To gain insight into molecular characterization, we investigated the differences in gene mutation between two risk score groups and found significant differences in the expression of *ARID1A*, *MYCBP2*, *AFF2*, *C3*. Furthermore, the low-risk group had a higher rate of *PCDH17* than the high-risk group. Loss of *ARID1A* function can lead to more aggressive tumor behavior and plays a key role in DNA repair and chromatin remodeling, meaning its mutation can promote tumorigenesis [42,43]. *MYCBP*2, *AFF2*, and *C3* influenced patient outcomes by affecting tumor growth, metastasis, and treatment resistance [44,45,46]. Our results also revealed that the high-risk group had a higher level of EMT activity, that provided more insight into tumor biology in different risk groups. Therefore, the high-risk patients had poor prognosis compared to low-risk patients, consistent with our survival findings.

Subsequently, we discussed how the gene signature correlates with established predictive biomarkers for immunotherapy response. Lately, immunotherapy has surfaced as a hopeful approach for the treatment of GC patients. Various ICIs such as CTLA-4 and PD-1 have displayed negative regulatory roles in the function of T cell immune function [4]. Furthermore, treatment guidelines for GC include Nivolumab and Pembrolizumab, both PD-1 inhibitors known for activating the immune system to enhance tumor cell elimination [47]. Currently, PD-L1 expression and TMB are key existing predictive models widely utilized in clinical practice for GC, with high PD-L1 expression and high TMB generally correlating with better response rates [48,49]. However, it is also understood that no single biomarker is universally sufficient, and optimal cut-off values can vary, leading to objective response rates (ORR) that may reach up to 33% even with careful selection based on these markers [50]. To predict the response to ICI, we explored the relationship between the risk score and IPS, TIDE in GC patients. TIDE scores correlate with T cell dysfunction and exclusion in tumors, while IPS is determined by z-scores of gene expressions of representative cell types. Our analysis showed that individuals categorized in the low-risk group showcased reduced TIDE scores compared to those in the high-risk group. Conversely, patients with low-risk scores showed higher IPS levels than those with high-risk scores. This implied that individuals with a low-risk profile might potentially gain greater benefits from ICI therapy. The results of this study are consistent with those reported by Zhang et al. [16], which identified gene signature that related with immune system in GC. Therefore, the risk score represented a promising hypoxia-immune-related prognostic biomarker, capable of predicting both the efficacy of ICI therapy and the overall survival of patients with GC.

In a tumor, a normal fibroblast can transform into a cancer-associated fibroblast (CAFs) which secrete growth factors such as transforming growth factor-beta (TGF-β), epidermal growth factor (EGF) or remodel the extracellular matrix (ECM); therefore, impact in tumor growth, progression, and metastasis [51]. In this study, we revealed that only *SRPX* was specifically expressed in fibroblast and the hypoxia-immune-related gene signature was found to be enriched in this cell. This indicated that fibroblasts are not only adapting to hypoxic conditions but also influencing immune cells, potentially leading to an immune-suppressive environment that benefits tumor growth. It highlighted the importance of targeting the fibroblast–hypoxia axis as a potential therapeutic approach for improving outcomes in GC patients. *SRPX* (sushi repeat-containing protein X-linked) has been implicated a crucial role in cellular responses to hypoxia. Under normal conditions, *SRPX* is involved in various cellular processes, including cell adhesion and migration. During hypoxic conditions, *SRPX* expression is upregulated, contributing to cellular adaptation and survival. In GC, *SRPX* has been identified as a potential tumor marker, with its expression levels correlating with tumor progression and metastasis [52]. Additionally, *SRPX* is implicated in modulating the immune response, particularly in the context of TME, where it may influence immune cell infiltration and activity [53]. Its enrichment in fibroblasts suggested that fibroblasts may use *SRPX* to facilitate these processes, potentially contributing to the development and aggressiveness of TME. This also demonstrated potential new therapeutic strategies that focus on disrupting fibroblast activity within TME.

There are several notable strengths in our study. Firstly, the formulation and validation of this signature were conducted across multiple datasets, affirming its robustness and reliability. In a second aspect, our gene signature associated with hypoxia and the immune system serves as a prognostic indicator and demonstrates potential predictive capabilities for immunotherapy responses in individuals with GC. Additionally, the creation of a nomogram using this gene signature shows promise for aiding its clinical implementation and advancement. Our gene signature helps identify high-risk GC patients with shorter survival, enabling more aggressive or novel treatment strategies, closer monitoring, and optimized patient counseling. Its immune-related factors suggest potential for predicting immunotherapy response, guiding patient selection. This signature is also valuable for clinical trial design, allowing for better patient stratification and accelerating targeted drug development. Our study also offered deeper insights into TME, tumor biology and tumorigenesis in GC. Despite the valuable findings of our study, it is important to acknowledge a few limitations. Conducting in vivo tissue modeling experiments would be beneficial for further validation of our results and to ensure their applicability in real-world settings. To further validate the gene signature identified in our study, it is crucial to include additional patient cohorts in future research. Expanding the dataset will enhance the robustness of our findings and improve their generalizability across diverse populations. Furthermore, unclear impact across different GC subtypes and potential batch effects from dataset integration will be addressed in future studies.

## 5. Conclusions

Our research has discovered a new gene signature that combines hypoxia and immune response in GC, utilizing both bulk and single-cell RNA-sequencing. By analyzing gene expression patterns in TME, we gain insight into the complex relationship between hypoxia and immune response in GC. Our study also emphasizes the importance of considering the immune profile in GC patients and provides insights into potential differences between two cohorts of risk score. This gene signature has important implications for personalized medicine in GC treatment, allowing clinicians to predict patient prognosis accurately and tailor treatment plans accordingly. Future directions will focus on rigorous clinical validation of this signature and its integration with other established biomarkers, such as MSI status or PD-L1 expression, to further refine patient stratification and therapeutic guidance. Overall, our study contributes to GC research and highlights the potential of the hypoxia-immune-related gene signature as a valuable prognostic tool, paving the way for personalized and effective treatment strategies in GC.

## Figures and Tables

**Figure 1 cimb-47-00552-f001:**
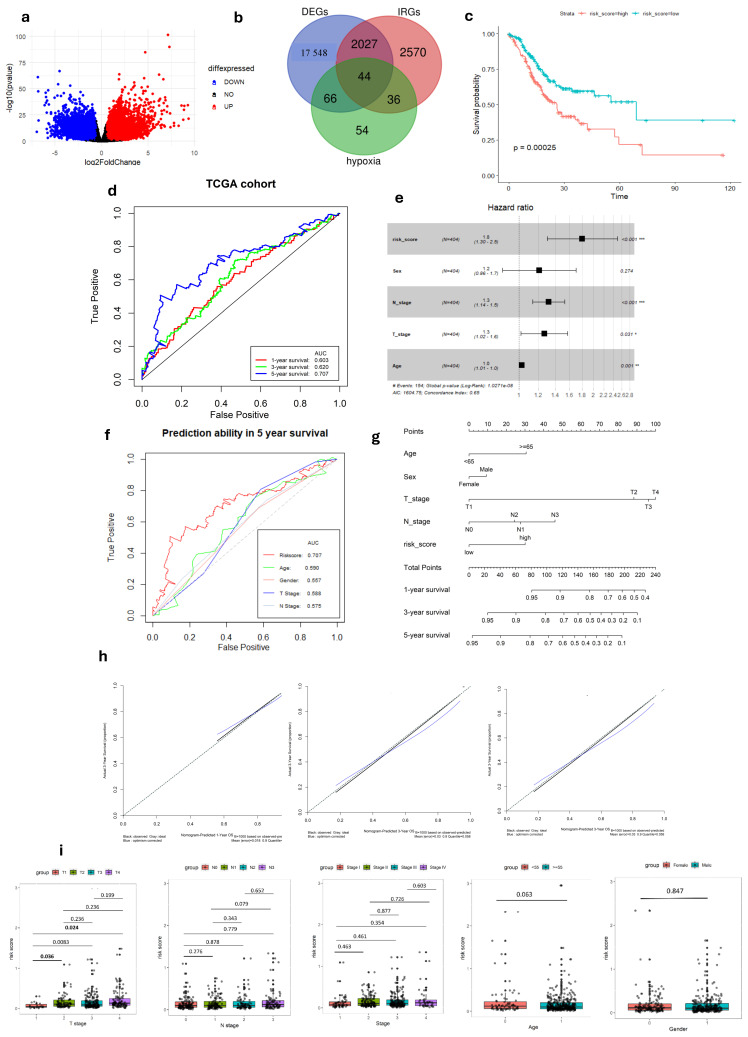
Development of Hypoxia-Immune-related gene signature in TCGA-STAD cohort. (**a**) Volcano plot of DEGs between tumor and normal GC tissue. (**b**) Venn diagram visualizing the intersection of three gene lists TCGA-STAD DEGs, Hypoxia-related genes and Immune-related genes. (**c**) K–M survival analysis of risk score. (**d**) ROC curve of predicting accuracy for 1, 3, and 5 years. (**e**) Forest plot of risk score and other clinical factors with *: *p* < 0.05, **: *p* < 0.01, ***: *p* < 0.001. (**f**) ROC analysis of risk score and clinical factors for overall survival at 5-year OS. (**g**) Nomogram for predicting 1, 3, and 5-year survival rates. (**h**) Calibration curves of nomogram. (**i**) Relationship of risk score and clinical factors including TMN stage, age, and gender.

**Figure 2 cimb-47-00552-f002:**
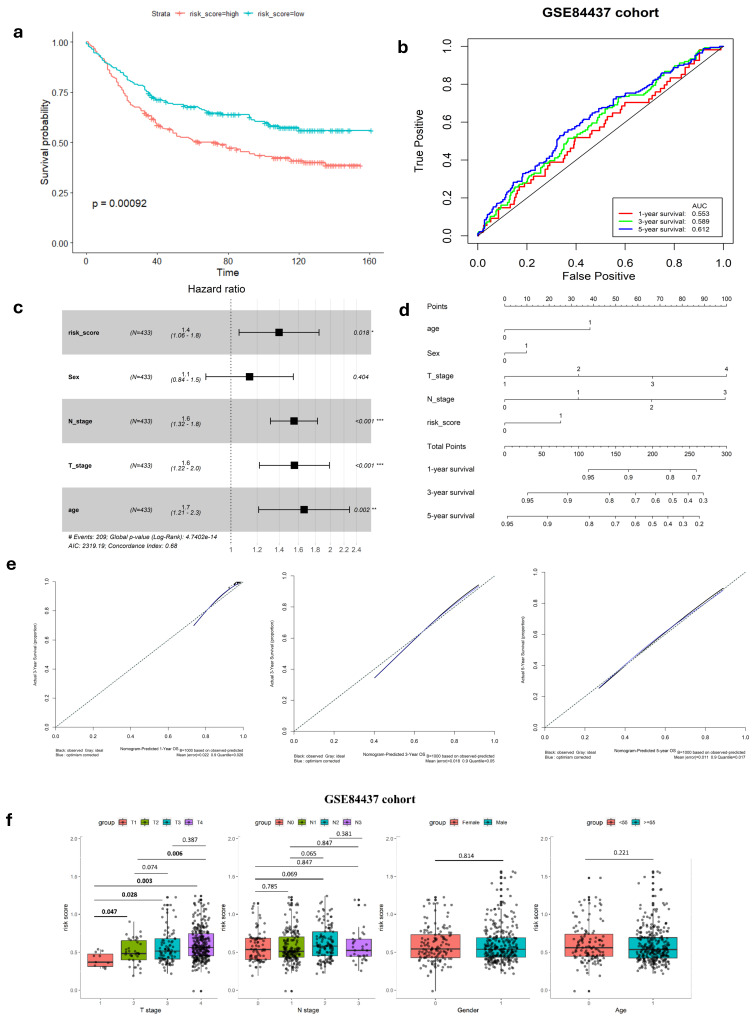
Validation of hypoxia-immune-related gene signature in GEO cohort. (**a**) K–M survival analysis of risk score. (**b**) ROC curve of predicting accuracy for 1, 3, and 5 years. (**c**) Forest plot of risk score and other clinical factors with *: *p* < 0.05, **: *p* < 0.01, ***: *p* < 0.001. (**d**) Nomogram for predicting 1, 3, and 5-year survival rates. (**e**) Calibration curves of nomogram. (**f**) Relationship of risk score clinical factors.

**Figure 3 cimb-47-00552-f003:**
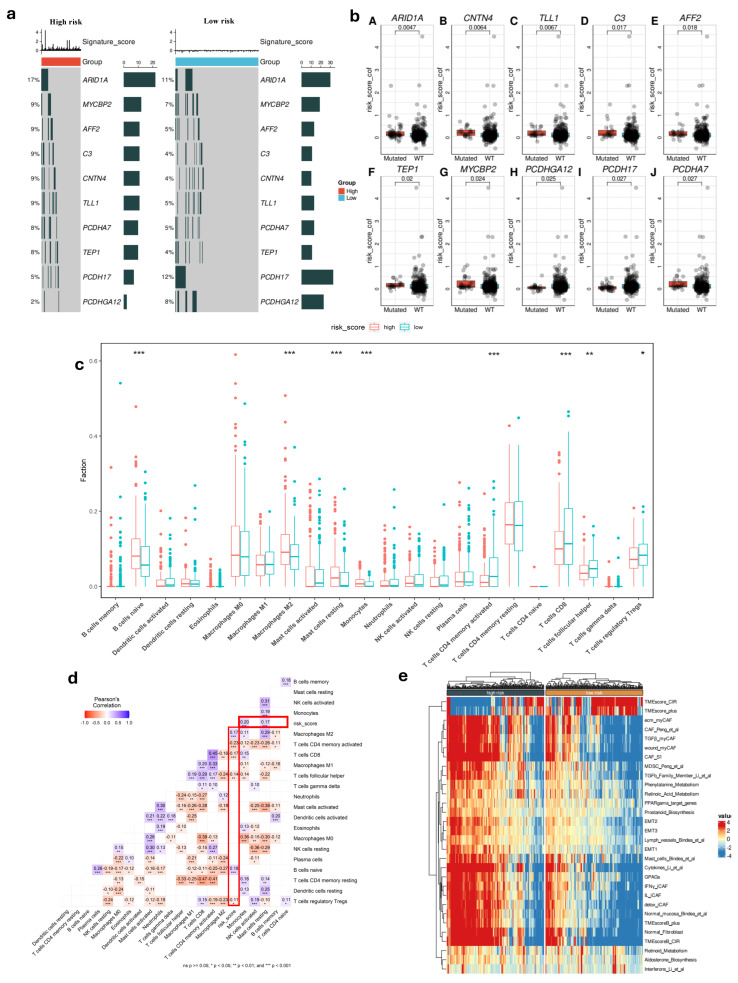
Molecular and immunological features in different risk score groups. (**a**) Top 10 gene mutation in two risk groups. (**b**) Difference between gene somatic mutations in risk score. (**c**) Distribution of 22 immune cells in low- and high-risk groups (*: *p* < 0.05, **: *p* < 0.01, ***: *p* < 0.001). (**d**) Correlation of risk score and immune cells. (**e**) Heatmap of biological signatures and pathway activity in high-risk and low-risk groups in TCGA cohort.

**Figure 4 cimb-47-00552-f004:**
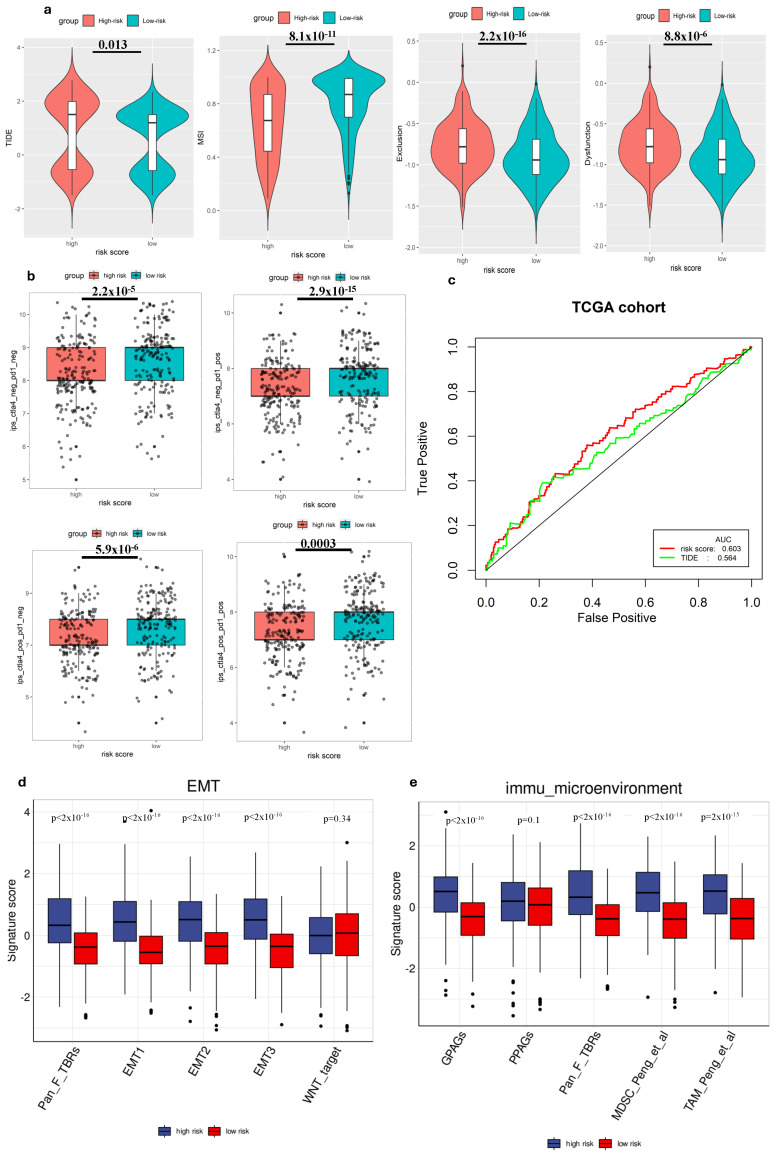
The ability to predict immune cell infiltration immunotherapy response of the gene signature. (**a**) Comparison of TIDE scores in different groups. (**b**) Comparison of IPS scores in different groups. (**c**) ROC analysis of risk score and TIDE for overall survival at 1-, 3-, and 5-year OS in the TCGA cohort. (**d**) Comparison of EMT-related signature scores between high-risk and low-risk groups. (**e**) Comparison of immune microenvironment signature scores between high-risk and low-risk groups.

**Figure 5 cimb-47-00552-f005:**
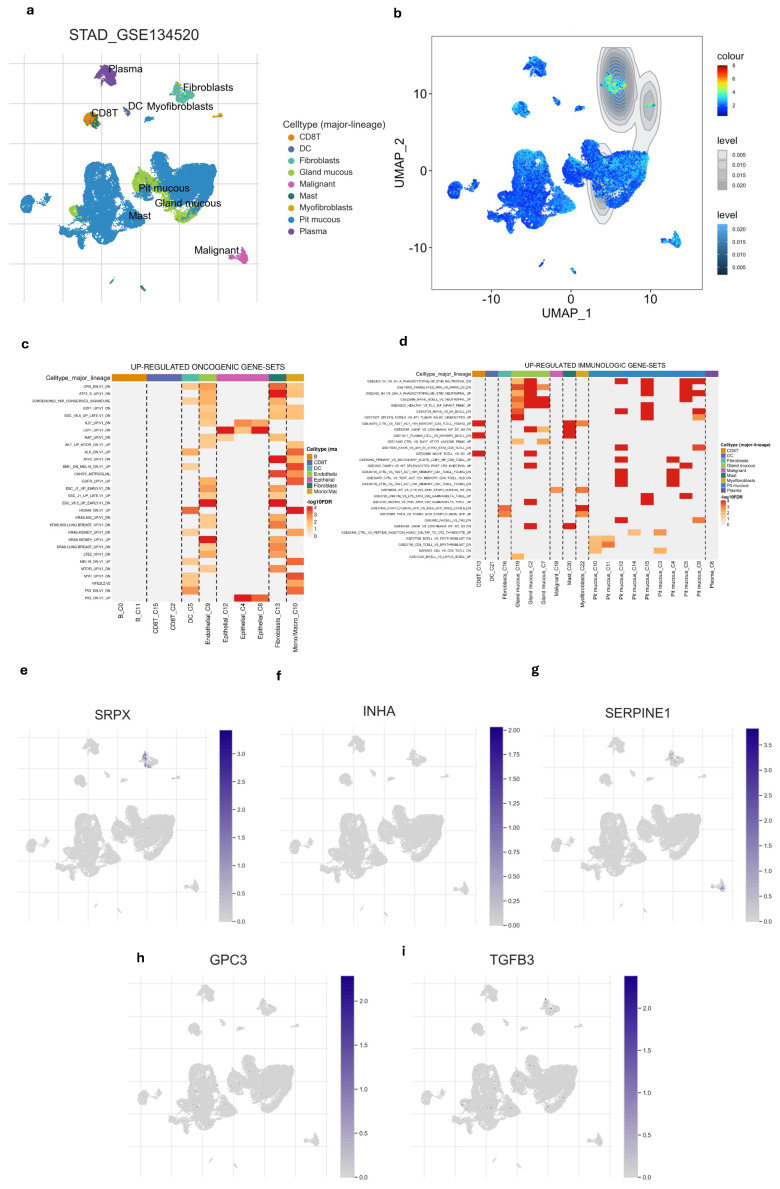
Single-cell profiling of gene signature. (**a**) UMAP of the 9 cell types. (**b**) the gene signature was highly enriched in fibroblasts compared to other cell types. (**c**) Heatmap of up-regulated oncogenic gene sets across major cell lineages. (**d**) Heatmap of up-regulated immunologic gene sets major cell lineages. (**e**–**i**) UMAP of every single gene in gene signature showed that SRPX is enriched in fibroblasts.

**Table 1 cimb-47-00552-t001:** Cox Regression Analysis of Clinicopathological Features and Risk Score Impact on OS in TCGA-STAD and GSE84437 Cohorts.

Factors	Univariate Analysis			Multivariate Analysis		
	HR	95%CI	*p*-Value	HR	95%CI	*p*-Value
TCGA cohort						
Age	1.52	1.094–2.113	0.0125 *	1.773	1.2707–2.475	0.000754 ***
Gender (male vs. female)	1.232	0.8774–1.73	0.228	1.201	0.8532–1.689	0.294109
T stage	1.306	1.072–1.591	0.00797 **	1.238	1.0004–1.533	0.049602 *
N stage	1.379	1.196–1.59	9.78 × 10^−6^ ***	1.312	1.1312–1.521	0.000328 ***
Risk score (high vs. low)	1.818	1.314–2.514	0.000305 ***	1.854	1.3381–2.569	0.000208 ***
GSE84437 cohort						
Age	1.02	1.007–1.032	0.00225 **	1.023	1.0108–1.036	0.000244 ***
Gender (male vs. female)	1.256	0.9275–1.7	0.141	1.184	0.8726–1.606	0.278218
T stage	1.74	1.378–2.198	3.31 × 10^−6^ ***	1.583	1.2385–2.024	0.000246 ***
N stage	1.676	1.429–1.967	2.42 × 10^−10^ ***	1.515	1.2871–1.782	5.72 × 10^−7^ ***
Risk score (high vs. low)	1.589	1.206–2.094	0.00101 **	1.395	1.0562–1.843	0.019045 *

* *p* < 0.05, ** *p* < 0.01, *** *p* < 0.001.

**Table 2 cimb-47-00552-t002:** The correlation between clinicopathological factors and the risk score.

	TCGA Cohort				GSE84437 Cohort			
	Total (N = 404)	Low Risk (N = 202)	High Risk (N = 202)	*p*-Value	Total (N = 433)	High Risk (N = 216)	Low Risk (N = 217)	*p*-Value
Gender								
Male	261 (64.6%)	134 (66.3%)	127 (62.9%)	0.5325	296 (68.4%)	147 (68.1%)	149 (68.7%)	0.892
Female	143 (35.4%)	68 (33.7%)	75 (37.1%)		137 (31.6%)	69 (31.9%)	68 (31.3%)	
Age								
<55	67 (16.6%)	30 (14.9%)	37 (18.2%)	0.4222	129 (28.9%)	69 (31.9%)	60 (27.6%)	0.329
≥55	337 (83.4%)	172 (85.1%)	165 (81.7%)		304 (70.2%)	147 (68.1%)	157 (72.4%)	
N stage								
N0	125 (30.9%)	66 (32.7%)	59 (29.2%)	0.4545	80 (18.5%)	39 (18.1%)	41 (18.9%)	0.011 *
N1	114 (28.2%)	59 (29.2%)	55 (27.2%)		188 (43.4%)	80 (37%)	108 (49.8%)	
N2	82 (20.3%)	42 (20.8%)	40 (19.8%)		132 (30.5%)	81 (37.5%)	51 (23.5%)	
N3	83 (20.5%)	35 (17.3%)	48 (23.8%)		33 (7.6%)	16 (7.4%)	17 (7.8%)	
T stage								
T1	22 (5.4%)	16 (7.9%)	6 (3%)	0.0222 *	11 (2.5%)	1 (0.5%)	10 (4.6%)	0.001 ***
T2	90 (22.3%)	41 (20.3%)	49 (24.3%)		38 (8.8%)	13 (6.0%)	25 (11.5%)	
T3	182 (45%)	99 (49%)	83 (41.1%)		92 (21.2%)	40 (18.5%)	52 (24%)	
T4	110 (27.2%)	46 (22.8%)	64 (31.7%)		292 (67.4%)	162 (75%)	130 (59.9%)	

* *p* < 0.05, *** *p* < 0.001.

## Data Availability

Data is contained within the article and Appendix A.

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
