# Peer review of "A Novel Hypoxia-Immune Signature for Gastric Cancer Prognosis and Immunotherapy: Insights from Bulk and Single-Cell RNA-Seq"

_cimb, 2025, doi:10.3390/cimb47070552_

Round 1

Reviewer 1 Report

Comments and Suggestions for Authors

The article under review is devoted to the study of hypoxia and its relationship with the immune response in gastric cancer. The authors propose a new gene signature that can predict the prognosis and response to immunotherapy in patients with this disease. The relevance of the study is due to the high mortality rate from gastric cancer and the need to improve the effectiveness of treatment. Hypoxia, or low oxygen levels in tissues, is a well-known factor that contributes to the progression of many types of cancer, including gastric cancer. It activates various molecular pathways that promote tumor growth and resistance to therapy.

The authors of the article conducted a comprehensive analysis of gene expression data obtained from bulk and single-cell RNA sequencing. This approach allowed them to identify a group of genes that are associated with hypoxia and the immune response. The resulting gene signature can be used to predict the prognosis of patients with gastric cancer and their response to immunotherapy.

The significance of hypoxia in activating tumor immunogenicity is an important aspect of the study. Hypoxia can affect the immune response by changing the expression of genes involved in antigen presentation, T cell activation and other processes. The authors of the article showed that the proposed gene signature takes into account these changes and can be used to optimize treatment strategies. The results of the study are of great importance for the formation of a response to antitumor therapy. The identification of patients who are likely to respond to immunotherapy is an important step in the personalized approach to treatment. The proposed gene signature can help clinicians make more informed decisions about the choice of therapy and improve the outcomes for patients 

Author Response

Comment 1: The article under review is devoted to the study of hypoxia and its relationship with the immune response in gastric cancer. The authors propose a new gene signature that can predict the prognosis and response to immunotherapy in patients with this disease. The relevance of the study is due to the high mortality rate from gastric cancer and the need to improve the effectiveness of treatment. Hypoxia, or low oxygen levels in tissues, is a well-known factor that contributes to the progression of many types of cancer, including gastric cancer. It activates various molecular pathways that promote tumor growth and resistance to therapy.

The authors of the article conducted a comprehensive analysis of gene expression data obtained from bulk and single-cell RNA sequencing. This approach allowed them to identify a group of genes that are associated with hypoxia and the immune response. The resulting gene signature can be used to predict the prognosis of patients with gastric cancer and their response to immunotherapy.

The significance of hypoxia in activating tumor immunogenicity is an important aspect of the study. Hypoxia can affect the immune response by changing the expression of genes involved in antigen presentation, T cell activation and other processes. The authors of the article showed that the proposed gene signature takes into account these changes and can be used to optimize treatment strategies. The results of the study are of great importance for the formation of a response to antitumor therapy. The identification of patients who are likely to respond to immunotherapy is an important step in the personalized approach to treatment. The proposed gene signature can help clinicians make more informed decisions about the choice of therapy and improve the outcomes for patients 

Response 1: Thank you for your highly insightful and detailed feedback on our manuscript. We're delighted you recognized the core significance of our work on hypoxia and its link to the immune response in gastric cancer. Your observations truly validate our efforts.

We agree that our use of both bulk and single-cell RNA sequencing was crucial, enabling us to develop a robust gen signature that we believe holds significant promise for prognosis prediction and immunotherapy response in the clinic.

Your emphasis on hypoxia's role in activating tumor immunogenicity is particularly appreciated, as this was a central focus of our investigation. We're confident our gene signature's ability to capture these complex changes will prove invaluable in optimizing personalized treatment strategies.

Reviewer 2 Report

Comments and Suggestions for Authors

Dear Authors,

First of all, congratulations for your interesting work. I hope that my hints will help you in the next steps of improvement and the final manuscript will be really valuable for the readers. It is very interesting and I hope it will be really good at the end of this process. 

There are several punctation mistakes (such as double space, double dot or no at all) and some typos - even if they do not change the value of the manuscript, I'd like to urge you to correct these imperfections. Also, several sentences look like they have been taken straight from the translator - grammar should be corrected. 

It might be a good idea to rethink the title. Current one is very long and not catchy, therefore it may not attract many potential readers, interested in the topic. Try to make it shorter and attractive. 

Not every abbreviation used has been explained, correct please.

Figures 1, 2, 3. elements of these figures are definitely too small, cannot be read not understood. 

Moreover, gene names should be written in italics, in opposite to the protein names, according to the rules of genetic consensus. Please, familiarise yourself with the rules and change the manuscript accordingly. Examples of rules summary can be found on websites such as: https://www.gmb.org.br/geneprotein-nomenclature-guidelinesor https://academic.oup.com/molehr/pages/Gene_And_Protein_Nomenclature

Can you think about clinical application of your discovery? It would be a good idea to add a paragraph about the possible usage, maybe there is a potential of having hospital-based rapid test? How it can be used, where it might be useful? 

Finally, I would like to encourage you to come back to the abstract part of your manuscript. In a current form it looks like a conference note or an abstract of a poster, it is not encouraging readers to dig deeper into your research, it seems not interesting. An abstract should have more popular-science style, to show the importance and significance of your work, not solely a mini-summary of what you have done. I strongly encourage you to rewrite this section.

Reviewer 3 Report

Comments and Suggestions for Authors

This manuscript titled “Development and validation of novel hypoxia-immune-related gene signature for predicting prognosis and immunotherapy response in gastric cancer on bulk and single-cell RNA-seq” presents a comprehensive bioinformatics study aimed at constructing and validating a prognostic gene signature for gastric cancer (GC). The authors integrate transcriptomic data from both bulk and single-cell RNA-sequencing with hypoxia- and immune-related gene sets to establish a five-gene signature predictive of patient survival and response to immune checkpoint inhibitors. The topic is timely and relevant, given the increasing importance of personalized immunotherapy in oncology. The manuscript is generally well-structured and methodologically sound; however, several areas require clarification, and more nuanced discussion to improve the rigor and translational impact of the study.

Abstract

  1. Consider emphasizing that the model was validated on both bulk and single-cell RNA-seq more explicitly to highlight the novelty.

Introduction

    1.  “especially immunotherapy which is a particularly expensive treatment...” This could be rephrased more scientifically to avoid informal tone. Suggest: “especially given the high cost of immunotherapy, particularly in low-resource settings.”
    2. Grammatical issue: “Immune cells has crucial role...” should be “Immune cells have a crucial role...”
    3. Consider condensing this paragraph. It repeats concepts about gene signature development without adding new background.

Methods

  1. Please clarify how genes from “hypoxia and immune-related genes” were intersected—were they filtered for DEGs first or after?
  2. “value of 50%” for immune cell inclusion is ambiguous. Do you mean “present in at least 50% of patients”? Please specify.
  3. URL for TIDE tool is broken (http://tide.dfci.atherard.edu/) — likely a typo; should be corrected to http://tide.dfci.harvard.edu/.

Results

  1. The AUC values (0.603–0.707) for survival prediction are moderate. Please discuss their practical clinical utility.
  2. Statistical significance is shown, but effect sizes (e.g., HRs) could be emphasized more in the main text.
  3. The authors describe “immune suppression” in high-risk patients. Please clarify which immune checkpoints or suppressive cells are enriched.
  4. The term “more normal tissue characteristics” is vague. Suggest rewording to “retained features of non-malignant tissue, including lower fibroblast infiltration and reduced EMT activation.”
  5. Figures 1–5: Effective in summarizing the data but ensure consistency of color coding between high-risk and low-risk groups across figures.

Discussion

  1. Rephrase “we discovered 44 hypoxia-immune-related genes, in which has strong association with...”. Suggested: “...including 44 genes significantly associated with hypoxia and immune pathways, particularly enriched in regulation of cell death and HIF-1 signaling.”
  2. Strong summary of immunotherapy response prediction. Could be improved by comparing with other existing predictive models (e.g., PD-L1, TMB).
  3. The discussion on SRPX and fibroblasts is valuable. However, it may benefit from clarification on whether SRPX is causally linked to prognosis or only correlative.
  4. The study’s limitations should be elaborated. For example:
      • Lack of prospective clinical validation.
      • Unclear impact across different GC subtypes (intestinal vs diffuse).
      • Potential batch effects from dataset integration.

Conclusion

  1. Consider adding a statement about the future direction: e.g., clinical validation, integration with other biomarkers like MSI status or PD-L1.
  2. Should specify actual dataset accession numbers rather than placeholder text.

Round 2

Reviewer 3 Report

Comments and Suggestions for Authors

I am satisfied with the authors’ responses to my previous comments and appreciate the revisions they have made to address the concerns raised. The manuscript is now significantly improved, and I believe it meets the standards for publication. I have no further concerns, and I recommend the paper for acceptance in its current form.